# Gastric examination using a novel three-dimensional magnetically assisted capsule endoscope and a hand-held magnetic controller: A porcine model study

Dong Jun Oh[ID]1ᵒ, Ji Hyung Nam1ᵒ, Junseok Park[ID]2, Youngbae Hwang3, Yun Jeong Lim[ID]1*

1 Department of Internal Medicine, Dongguk University College of Medicine, Dongguk University Ilsan Hospital, Goyang, Republic of Korea, 2 Digestive Disease Center, Institute for Digestive Research, Department of Internal Medicine, Soonchunhyang University College of Medicine, Seoul, Republic of Korea, 3 Department of Electronics Engineering, Chungbuk National University, Cheongju, Republic of Korea

ᵒ These authors contributed equally to this work.
* drlimyj@gmail.com

**Data Availability Statement:** The research submitted to PLOS ONE (PONE-D-20-31353R2) was conducted jointly with Dongguk University

## Abstract

Magnetically assisted capsule endoscopy (MACE) is a noninvasive procedure and can overcome passive capsule movement that limits gastric examination. MACE has been studied in many trials as an alternative to upper endoscopy. However, to increase diagnostic accuracy of various gastric lesions, MACE should be able to provide stereoscopic, clear images and to measure the size of a lesion. So, we conducted the animal experiment using a novel three-dimensional (3D) MACE and a new hand-held magnetic controller for gastric examination. The purpose of this study is to assess the performance and safety of 3D MACE and hand-held magnetic controller through the animal experiment. Subsequently, via the dedicated viewer, we evaluate whether 3D reconstruction images and clear images can be obtained and accurate lesion size can be measured. During real-time gastric examination, the maneuverability and visualization of 3D MACE were adequate. A polypoid mass lesion was incidentally observed at the lesser curvature side of the prepyloric antrum. The mass lesion was estimated to be 10.9 x 11.5 mm in the dedicated viewer, nearly the same size and shape as confirmed by upper endoscopy and postmortem examination. Also, 3D and clear images of the lesion were successfully reconstructed. This animal experiment demonstrates the accuracy and safety of 3D MACE. Further clinical studies are warranted to confirm the feasibility of 3D MACE for human gastric examination.

## Introduction

Capsule endoscopy (CE) was first introduced in 2001 [1]. Recent guidelines recommend CE as the first line method for identification of small bowel lesions or obscure gastrointestinal (GI) bleeding because CE is non-invasive and allows direct visualization of the small bowel mucosa

Ilsan Hospital (Prof. Yun Jeong Lim). The data sets (magnetic controller, magnetic 3D capsule endoscopy, Miroview software) are owned by Intromedic Company. There are no legal or ethical restrictions. We permit the use of data in research article. No author has privileged access to the data set. Contact information: Email: kskim@intromedic.com Phone: +82-10-2691-0334.

**Funding:** This experiment was supported by a grant from the Korean Health Technology R&D project through the Korean Health Industry Development Institute (KHIDI), funded by the Ministry of Health & Welfare, Republic of Korea (Grant number: HI19C0665). The corresponding author, Dr. LYJ, received the fund. The funders did not play a direct role in writing this article, but this study could be constructed on the fund.

**Competing interests:** The authors have declared that no competing interests exist.

[2, 3]. Along with the development of high frame rates for CE camera, some studies reported the use of CE for the evaluation of esophagus and colon (small bowel CE at 2–6 frame rate, esophagus and colon CE at 18–35 frame rate) [1, 4, 5]. However, the entire gastric examination is limited because the stomach has a large diameter and volume compared to other parts of the GI tract. And CE passes the proximal gastric region rapidly and images obtained by passive CE movement are only visible [6]. The first study of human gastric examination by utilizing CE with magnetic force was published in 2010 [7], and several studies have reported favorable outcomes of entire gastric examination using CE with a magnetic controller [8–11]. A few studies have questioned the effectiveness of gastric examination using CE with a magnetic controller [12, 13]. Although several studies were shown the inconsistent findings, gastric examination using a magnetic controller is remarkable in that it can overcome passive movement, which is a major problem of conventional CE. In addition, it is associated with minimal procedural discomfort, is non-invasive, and does not require sedation, as does conventional CE [6, 8]. CE with a magnetic controller has been applied to screen for gastric cancer in China [9], but it is still necessary to improve the quality of CE images. As gastric cancer appears in various forms [14], accurate identification of a gastric lesion may be difficult with two-dimensional (2D) CE images. In order to reproduce high-quality images, additional technologies including three-dimensional (3D) reconstruction of the image, improvement of image sharpness, and size measurement are required. In addition, the robot magnetic control system reported in several studies [9, 10] is bulky and has a hurdle associated with installation [11]. The hand-held magnetic controller developed by Intromedic Co. Ltd (Seoul, Korea) [11, 15] can overcome this hurdle. However, it is still inconvenient to operate. So, it is necessary to develop a novel magnetic controller that is more user-friendly.

Thus, in this animal experiment, we used a novel 3D magnetically assisted capsule endoscopy (MACE), a hand-held magnetic controller, and dedicated viewer to identify whether we can achieve higher quality images and better performance than are possible with a conventional MACE system.

## Methods

### 1. Subject

A mini-pig (male, 34.8 kg, *Sus scrofa*, OPTIPHARM Co. Ltd, Chungju, Korea) was used for this experiment. On the day of the experiment, a veterinarian administered zolazepam and xylazine via intramuscular injection (3.48 mL, 0.1 mL/kg) and administered general anesthesia (isoflurane). Laboratory animal breeding and experiment were carried out at animal experiment facility (KNOTUS Co. Ltd, Incheon, Korea) registered with Korea Food & Drug Administration. The study protocol was approved by The Animal Care and Use Committee of KNOTUS IACUC (the Institutional Animal Care and Use Committee at the KNOTUS Co. Ltd), approval number KNOTUS IACUC 20-KE-145. All endoscopic procedures were performed by three specialized endoscopists at Dongguk University Ilsan Hospital (DJO, JHN, and YJL).

### 2. Devices

Upper endoscope with monitoring system (EVIS Lucera CV 260SL, Olympus Co. Ltd, Japan), 3D MACE (MiroCam® MC 4000-M, Intromedic Co. Ltd, Korea) with a hand-held magnetic controller, a real-time receiver with sensor belt (MR 2000, Intromedic Co. Ltd, Korea) (Fig 1), and dedicated 3D MACE viewer (Miroview® MC 4000, Intromedic Co. Ltd, Korea) were used.

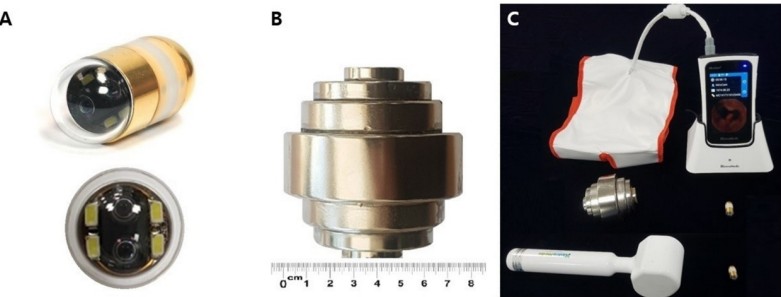

**Fig 1. Overview of the three-dimensional (3D) magnetically assisted capsule endoscopy (MACE) system.** A: 3D MACE (MiroCam® MC 4000-M), B: Circular stacked magnetic controller, C: Real-time receiver with sensor belt (MR 2000). The size of the existing controller and the new controller was compared.

**(1) 3D MACE.** Unlike conventional MACE, the 3D MACE is equipped with dual stereo cameras. This dual stereo camera enables size measurement and image reconstruction. The size and weight of the 3D MACE is slightly longer and heavier than those of the conventional MACE (11 x 25.5 mm, 4.5 g in 3D MACE versus 11 x 24.5 mm, 4.2 g in MACE) (S1 Fig). Except for the size and dual stereo cameras, the basic properties of 3D MACE are the same as that of conventional MACE [11]. The image resolution is 320 x 320 pixels and the visual field is 170 degrees. The total battery time is 8 hours and the record rate is 2 frame per second each camera.

**(2) Hand-held magnetic controller.** In this study, a new circular stacked magnetic controller (7 cm in length and 6.4cm in width) was used. When the side of the controller is rotated, the 3D MACE rotates 360˚, and when tilted towards the tip of the controller, the camera portion of 3D MACE moves up and down (S2 Fig). The new controller is smaller in size than an existing controller (hammer shape, 26 cm in length, 6.5cm in hammer head width) [11]. But new controller is equipped with a higher-grade magnet than the magnet of existing controller (N54 versus N35). So, the magnetic force measured 10cm away from the controller was similar in the new and existing controller. (30 gauss in new controller and 27 Gauss in existing one, respectively). The magnetic force with respect to the MACE by the controller was measured to be 41.2 mN at a distance of 10.5 cm. The effective operation distance between the magnetic controller and 3D MACE is up to 12.5cm.

**(3) Real-time receiver with sensor belt.** Real time receiver uses Android version 9, with an LCD resolution of 800 x 480 pixels. The receiver and sensor belt can be connected. The pads of the sensor belt are composed of 9 conductive fibers (S3 Fig). The images recorded by the 3D MACE are delivered to the real time receiver using a novel transmission technology, known as electrical field propagation. This technology has been applied from the existing small bowel capsule endoscope of the Intromedic Co. Ltd [16].

**(4) The dedicated viewer.** We used a new dedicated viewer (S4 Fig) equipped with 3D rendering and size measurement algorithms (Fig 2). Using the parallax of the left and right images acquired from the dual 3D MACE stereo cameras, the actual-scale depth information for each pixel is computed by finding stereo correspondences using the direct attenuation model [17]. Then, the size of the target lesion is measured based on the depth information. The 3D model can be visualized by 3D rendering reconstructed depth information with texture mapping of a color image [18]. In addition, using deep learning-based image processing with de-noising and de-blurring, the image sharpness can be increased. The automated gain control is used to improve brightness of images.

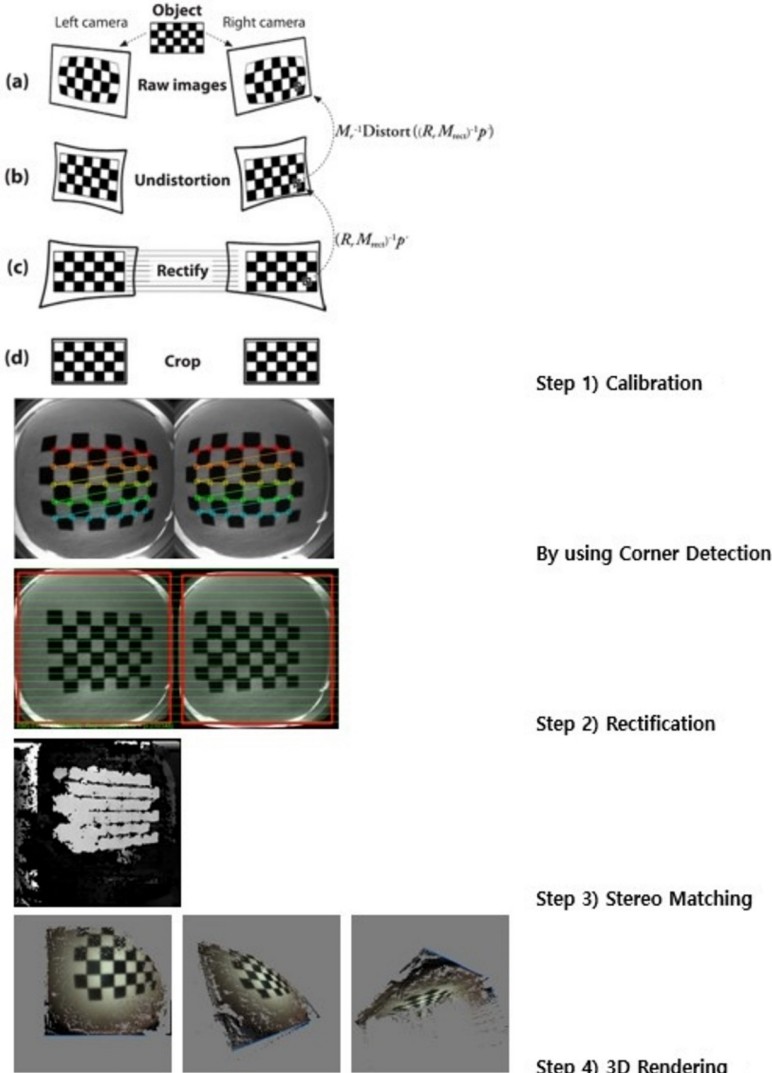

**Fig 2. The process of three-dimension (3D) reconstruction in 3D magnetically assisted capsule endoscope and dedicated viewer.**

## 3. Porcine gastric examination using the 3D MACE and a hand-held magnetic controller

A sensor belt was attached to the upper abdomen of the mini-pig. At the left lateral position, an upper endoscope was inserted into the mini-pig and the 3D MACE was dropped onto the stomach using an endoscopic net. Subsequently, the pig's posture was changed to the supine position, and a magnetic controller was placed on the sensor belt. Under upper endoscopic monitoring, consistent movement of the magnetic controller and 3D MACE was confirmed in real time. Then, normal saline (800 cc) mixed with simethicone (40 mg, TaeJoon Pharm Co. Ltd, Seoul, Korea) was administered to the stomach through the upper endoscope in order to fill the stomach without forming air bubbles. When the stomach was filled with normal saline, the upper endoscope was removed.

The 3D MACE was moved from the fundus to the antrum with rotation and tilting. For reconfirmation of the stomach, the 3D MACE was moved back from the antrum to the fundus. The quality of the images acquired while moving the 3D MACE was identified with a real-time receiver. When a gastric lesion was suspected in the real-time receiver, the characteristics of the lesion, such as location, shape, and size, were determined. And the upper endoscope was reinserted for assessment of validation and safety. We identified whether the images obtained from the real-time receiver of 3D MACE matches the image obtained from upper endoscopy.

Following the animal experiment, the 3D MACE images were reviewed and reorganized by a dedicated viewer. Size measurement, 3D reconstruction, and improvement of image sharpness were achieved with the dedicated viewer through some sophisticated steps.

## 4. Safety

After the gastric experiment was over, the 3D MACE was moved to the duodenum. And upper endoscopy and chest radiography were used to confirm any complications, such as bleeding, perforation, or aspiration. Until the 3D MACE was released from anus, the subject was properly managed and observed in the animal experiment facility for assessing the capsule transit time and any other complications.

## Results

### 1. Porcine gastric examination using 3D MACE and a hand-held magnetic controller

When the 3D MACE was dropped from the esophagogastric junction, it was located in the gastric fundus. The maneuverability of 3D MACE with a hand-held magnetic controller was confirmed by upper endoscopy (S1 Video). The image obtained by 3D MACE was also confirmed with a real-time receiver, and upper endoscopy was withdrawn. Gastric examination was performed while inspecting the fundus, body and antrum with various manipulations (S5 Fig). Incidentally, at the antrum, a protruding mass lesion was observed on the lesser curvature side of the pre-pyloric region. So, 3D MACE was manipulated and observed from various angles to identify the mass lesion. We tried to transit the 3D MACE into the duodenum but it failed to pass the pylorus that had been narrowed by lesions. Instead, we were able to acquire some images of the mass lesion that appeared to originate at the duodenal bulb. No additional lesion was observed until the end of the examination. Gastric examination with the 3D MACE lasted about 30 minutes. Re-examination of the stomach via upper endoscopy revealed no specific lesion except for the mass lesion at the pre-pyloric antrum. The shape and location of the mass lesion was the same as when inspected by the real-time receiver. The size of the mass lesion as measured by an endoscopic ruler was approximately 11 x 13 mm (length x width) (S6 Fig).

### 2. Verification of images of the gastric lesion by a dedicated viewer with size measurement, 3D reconstruction and image enhancement

The recorded images from the gastric examination were uploaded and analyzed with a dedicated viewer. The gastric landmarks and the mass lesion were readily identified with the dedicated viewer (Fig 3). The size of the mass lesion as determined with the dedicated viewer was 10.9 x 11.5 mm (length x width) (Fig 4A), which resembled with the size as measured by upper endoscopy. The 3D images of the mass lesion were reconstructed, and the shape of the mass lesion was accurately represented (Fig 4B). The sharpness of image was obtained through enhanced image processing (Fig 4C).

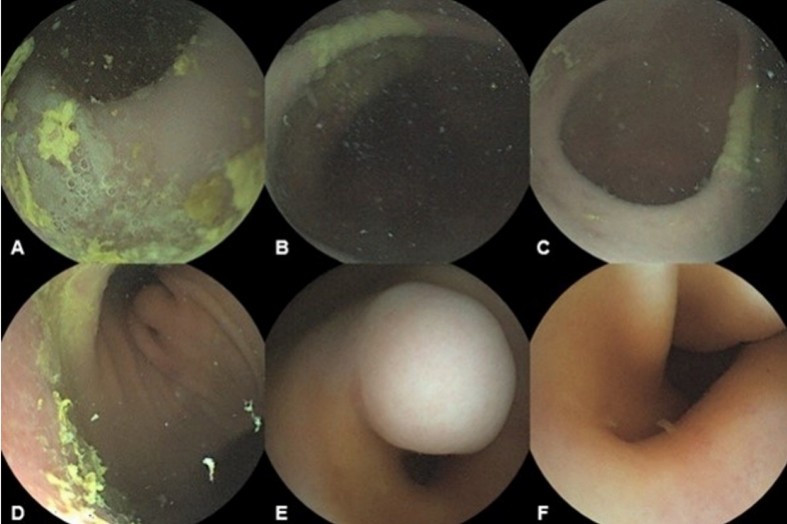

**Fig 3. The dedicated viewer images of porcine stomach taken via three-dimensional magnetically assisted capsule endoscopy.** A: Fundus, B: Cardia to high body, C: Midbody to gastric angle, D: Low body to antrum, E: Prepyloric antrum, F: Duodenal bulb.

### 3. Safety

No significant adverse event was observed when reconfirmed by upper endoscopy. We moved the 3D MACE into the duodenum using the gastroscopic net. The chest radiograph also showed no abnormal findings. The 3D MACE was released from the mini-pig 7 days after the experiment, without any complications.

### 4. Confirmation of gastric lesion through autopsy

The release of the 3D capsule was confirmed, an autopsy was performed to confirm the gastric mass lesion in animal experiment facility. The subject was euthanized using potassium chloride (2 mEq/kg to effect, intravenous injection) under general anesthesia with isoflurane and unnecessary pain was minimized. The mass lesion was confirmed by dissecting the stomach up to the duodenal bulb. As analyzed by 3D MACE, the mass lesion originated in the duodenal bulb and measured about 11 mm x 12 mm (length x width) in size (Fig 5).

## Discussions

We conducted porcine gastric examination using a new 3D MACE and a hand-held magnetic controller. The new 3D MACE and magnetic controller was used to explore the entire stomach of the mini-pig and exactly delineate a gastric lesion. The device demonstrated successful maneuverability and safety in the experiment, and it enabled not only acquisition of 3D reconstruction and improved sharpness images, but also accurate size measurement.

CE is non-invasive and does not require sedation. It can be used for examination of the esophagus and colon [1, 19]. However, gastric examination using CE is limited because the large size of the stomach cavity and passive CE movement cause insufficient examination [12]. Therefore, active locomotion of CE was developed with a focus on internal locomotion; however, it was impossible to incorporate all of the necessary elements with a limited capsule size in addition to power limitations [6, 20]. So, the interest in external magnetic controls for CE, a more realistic option, has increased. Shortly after the first human gastric examination by

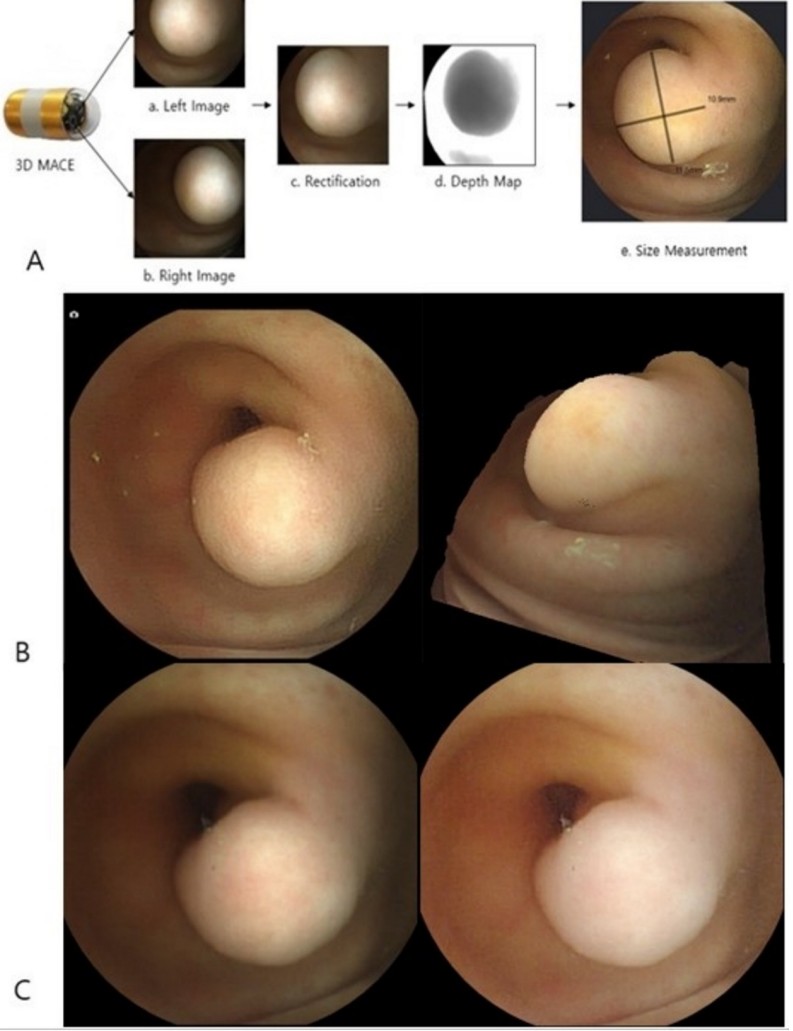

**Fig 4. Reconstructed images with the dedicated viewer.** A: The process of size measurement by three-dimensional (3D) magnetically assisted capsule endoscopy (MACE). B: The left is original image. The right is the 3D viewer mode and rotated 3D image. C: The left is original image. The right is mode of image enhancement. Compared to conventional 2D MACE, a clearer image was obtained by dedicated viewer.

magnetic CE manipulation [7], several clinical studies demonstrated the feasibility, efficacy, and safety of gastric examination by using magnetic CE manipulation [21–24].

Another important issue is whether CE can be used to accurately distinguish and measure lesions. Differences in color based on bleeding or ulceration are easy to detect with conventional 2D CE, whereas it can be difficult to distinguish uneven lesions, such as excavated or elevated lesions. Therefore, in an attempt to acquire 3D images with CE, the focus was initially on the development of 3D reconstruction software because hardware for 3D systems is limited by power consumption and packaging issues [25, 26]. Interestingly, the 3D reconstructed images improved the interpretation performance of novice CE readers, but not that of expert readers [27]. So, the availability of 3D image reconstruction for CE may increase diagnostic accuracy for some physicians. However, several studies reporting the feasibility of 3D software showed inconsistent findings [27–29]. A recently developed 3D CE (MiroCam® MC 4000, Intromedic

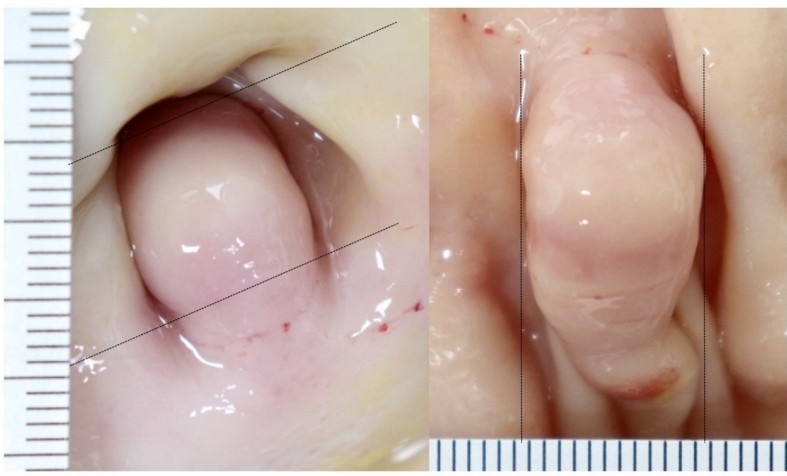

**Fig 5. Postmortem examination.** The mass is approximately 11 x 12 mm (length x width) in size and originates in the bulb.

Co. Ltd, Seoul, Korea) is the first hardware-enabled 3D reconstruction device to use stereo matching technology [30], and has been demonstrated in phantom models to estimate the size of an object within 10% relative error to its actual size [17, 31]. The 3D MACE presented in our study represents a new model providing additional magnetic control to the 3D CE. We obtained 3D images of the lesion more accurately via 3D MACE, which enabled precise measurement of lesion shape.

A conventional 2D MACE (MiroCam® MC 1000-WM, Intromedic Co. Ltd, Seoul, Korea) was examined in previous studies but used a hammer-shaped magnetic controller, unlike the controller used in our study [32]. After the 2D MACE was tested in an excised stomach model [33], it demonstrated good feasibility for the detection of gastric lesions in a clinical study [11]. Another study reported that the diagnostic accuracy of the 2D MACE was comparable to that of upper endoscopy for detection of bleeding foci in patients with suspected upper GI bleeding. In addition, it showed that detectability of focal gastric lesions on 2D MACE was higher than that on upper endoscopy, and the tolerability of 2D MACE was also superior [34]. 3D MACE used in our study can get higher-quality images than 2D MACE, but there are only a 0.3 g and 1 mm difference. The size of the mass lesion as measured with 3D CE was very similar to the size as confirmed by autopsy, and was even more accurate than the size measured with a gastroscopic ruler.

In a recent Korean study based on the 2,914 CE registry, the incomplete small bowel transit rate of CE was as high as 33% [35]. Even though it may be related to various factors such as age and bowel preparation [35], it is also affected by battery power limitations in CE. The impact of MACE on complete small bowel transit was assessed to examine whether magnetic steering reduced the gastric transit time of CE. But, magnetic steering with conventional MACE failed to overcome pyloric contractions at first [36]. Interestingly, we easily moved the 3D MACE to the pylorus using the new hand-held magnetic controller and observed the duodenal bulb from the pyloric ring. The mass lesion located in the prepyloric antrum was of duodenal origin, and narrowed the pyloric ring. If there was no mass lesion near the pylorus, the capsule would easily have passed through the pylorus.

The circular stacked magnetic controller has a smaller volume (ratio: 0.69 vs. 1) but a higher magnet grade (neodymium magnets: N54 vs. N35 per volume) [37] than the existing hammer-shape magnetic controller. Therefore, the measured magnetic forces of the two controllers are

similar. However, the hammer-shaped controller is difficult to handle because the contact surface with the abdominal wall is wide, whereas the stacked controller can be held with a single hand and bears a smaller contact surface with the abdominal wall, making it more effective for CE manipulation. Of course, the measured magnetic force of the new controller should be determined before clinical application.

Worldwide, gastric cancer is the fifth most common cancer and the third leading cause of cancer-related death. The incidence rate is especially high in eastern Asia, including Korea [38]. Therefore, biennial gastric cancer screening via either upper endoscopy or upper gastrointestinal series (UGIS) for individuals aged 40 or older has been conducted in Korea since 1999. However, UGIS is not effective at reducing cancer mortality [39]. Upper endoscopy, on the other hand, is an effective modality for reducing gastric cancer mortality; however, it is invasive and sedation-related complications can be fatal [40]. Gastric cancer screening of 34 individuals using magnetically controlled CE (MCE) with an automatic robotic control system (Ankon Technologies Co. Ltd, Shanghai, China) was first attempted in China [24]. Subsequently, several comparative studies showed satisfactory performance and diagnostic yield of MCE, comparable to upper endoscopy [10, 41]. In a recent large population-based cohort study, a robotic MCE system was considered safe and clinically feasible for gastric cancer screening [9]. However, the large volume of the robotic equipment remains a challenge. The robotic MCE system reduced gastric transit time [42], and our new hand-held controller could allow for pyloric passage. Clinical applications and technical improvements to the hand-held magnetic controllers are necessary to replace upper endoscopy with MACE for gastric cancer screening in the future.

Since MACE is still in the developmental stage, clinicians have limited experience with operating hand-held magnetic controllers, and clinical technique can influence the results of gastric examination. A protocol for the manipulation of hand-held magnetic controllers is needed, along with further clinical studies. Nevertheless, the present study was the first to evaluate magnetic control of 3D MACE and to verify the application of a new hand-held magnetic controller. Also, we demonstrated meticulous size measurement as well as detection of gastric lesion with the new 3D MACE system. This preclinical trial provides a basis for further clinical studies on 3D MACE. Furthermore, we suggest that 3D MACE with the new hand-held controller may be used for gastric cancer screening in high-risk areas for patients in whom upper endoscopy is contraindicated or not well tolerated. Further human studies and sufficient safety assessments should be conducted before clinical application of 3D MACE.

## Supporting information

**S1 Fig. Size comparison between conventional magnetically assisted capsule endoscope (MACE) and three-dimensional (3D) MACE.** The left is conventional MACE, and the right is 3D MACE. 3D MACE is about 1mm longer than conventional MACE.
(TIF)

**S2 Fig. Real-time movement in a gastric model of three-dimensional magnetically assisted capsule endoscope controlled by the magnetic controller.**
(TIF)

**S3 Fig. Sensor belt and real-time receiver.**
(TIF)

**S4 Fig. Real-time screen of dedicated viewer.** Three-dimension reconstruction of image and size measurement are possible with a new dedicated viewer.
(TIF)

**S5 Fig. Actual picture of animal experiment.**
(TIF)

**S6 Fig. Size measurement of mass lesion using an endoscopic ruler.**
(TIF)

**S1 Video. Confirming the movement of 3D MACE under gastroscopy.**
(AVI)

## Author Contributions

**Conceptualization:** Dong Jun Oh, Yun Jeong Lim.

**Data curation:** Junseok Park, Yun Jeong Lim.

**Formal analysis:** Ji Hyung Nam.

**Funding acquisition:** Yun Jeong Lim.

**Investigation:** Youngbae Hwang.

**Methodology:** Dong Jun Oh, Ji Hyung Nam.

**Project administration:** Yun Jeong Lim.

**Resources:** Dong Jun Oh, Yun Jeong Lim.

**Software:** Youngbae Hwang.

**Supervision:** Junseok Park, Yun Jeong Lim.

**Validation:** Junseok Park, Yun Jeong Lim.

**Visualization:** Ji Hyung Nam.

**Writing – original draft:** Dong Jun Oh, Ji Hyung Nam.

**Writing – review & editing:** Yun Jeong Lim.

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
