## [Decision Letter · Decision Letter 0]

3 Feb 2021

PONE-D-20-31353

Gastric examination using a novel three-dimensional magnetically assisted capsule endoscopy with a hand-held magnetic controller: A porcine model.

PLOS ONE

Dear Dr. LIM,

Thank you for submitting your manuscript to PLOS ONE. After careful consideration, we feel that it has merit but does not fully meet PLOS ONE’s publication criteria as it currently stands. Therefore, we invite you to submit a revised version of the manuscript that addresses the points raised during the review process.

We look forward to receiving your revised manuscript.

Kind regards,

Edoardo Sinibaldi

Academic Editor

PLOS ONE

Additional Editor Comments:

Dear Authors,

you are kindly encouraged to improve the manuscript by leveraging the points raised by the Reviewers, in particular by Reviewer #1.

You are also kindly invited to declare any potential conflicts of interest.

Best regards

Journal Requirements:

2. As part of your revisions, please specify whether or not you are reporting on a terminal surgery. Please state the method of euthanasia. If this was a survival surgery, please provide details about post-operative analgesics and relevant post-operative supportive care. We thank you for your attention in this matter.

"This research was supported by T2B Infrastructure Center for Digestive Disorders"

" This experiment was supported by a grant from the Korean Health Technology R&D project through the Korean Health Industry Development Institute (KHIDI), funded by the Ministry of Health & Welfare, Republic of Korea (Grant number: HI19C0665)."

We note that one or more of the authors are employed by a commercial company: Intromedic CO., LTD.

4.1. Please provide an amended Funding Statement declaring this commercial affiliation, as well as a statement regarding the Role of Funders in your study. If the funding organization did not play a role in the study design, data collection and analysis, decision to publish, or preparation of the manuscript and only provided financial support in the form of authors' salaries and/or research materials, please review your statements relating to the author contributions, and ensure you have specifically and accurately indicated the role(s) that these authors had in your study. You can update author roles in the Author Contributions section of the online submission form.

4.2. Please also provide an updated Competing Interests Statement declaring this commercial affiliation along with any other relevant declarations relating to employment, consultancy, patents, products in development, or marketed products, etc.  

Reviewers' comments:

Reviewer's Responses to Questions

**Comments to the Author**

1. Is the manuscript technically sound, and do the data support the conclusions?

Reviewer #1: Partly

Reviewer #2: Yes

2. Has the statistical analysis been performed appropriately and rigorously? 

Reviewer #1: N/A

Reviewer #2: N/A

3. Have the authors made all data underlying the findings in their manuscript fully available?

Reviewer #1: No

Reviewer #2: Yes

4. Is the manuscript presented in an intelligible fashion and written in standard English?

Reviewer #1: No

Reviewer #2: Yes

5. Review Comments to the Author

Reviewer #1: Dear Authors,

Thank you for reporting this interesting piece of work reporting the gastric examination of a porcine model using a 3D magnetically assisted capsule endoscopy system. Please find below my comments and questions concerning the various sections of your manuscript.

- INTRODUCTION -

1. Please quantify or provide a range in frames-per-sec on what the 'high' in

"development of high frame rate of CE camera" refer to.

- METHODS -

1. This first paragraph describes that "Laboratory animal breeding and experiment were carried out in strict accordance with ethical guidelines and carried out at KNOTUS Co. Ltd (Incheon, Korea), a certified facility." What are the ethical guidelines of KNOTUS? Please attach the guidelines as supplementary information or link it to a publically available URL.

2. Why was the study restricted to just one subject? Considering that the MACE was not able to pass the pylorus and travel through the small and large intestines because of the presence of the lesion, isn't this pre-clinical study preliminary and perhaps even incomplete to determine the efficacy of this 3D MACE and new magnetic controller?

3. Fig 1A, B, and C are generally of very low quality and appear not in focus. Please replace these images with higher quality images.

(a) - Please use a scale bar and mark the dimensions of the capsule in Fig1A.

(b) - The pictures of the capsule in Fig1A do not clearly show the 2 camera system that is important detail in this novel capsule. Please replace it with a macro-photograph that showcases the camera and imaging system on the capsule clearly.

(c) Fig C is also of very low quality. None of the text on display nor the label on the product is legible.

4. The functionality of the real-time receiver and sensor belt has to be described. What sensors are used in the belt? What kind of receivers on the real-time receiver. Please include a schematic of the whole 3D capsular endoscopy system.

5.

(a) In Section 2. Devices, Subsection (2) Hand-held magnetic controller, it is not clear what a conventional controller is? Is it an existing Intromedic product? What are its dimensions? How dos the new magnetic controller compare to other research-stage or other commercial handheld controllers? Please provide refernces to other controllers as well. What would be most appropriate is if controllers and their magnet configurations can be compared in a figure.

(b) It is mentioned that "the magnetic force is equal between the two"? How has this been calculated or estimated. The other sections of the paper do not provide satisfactory explanation to this.

- RESULTS -

1. In Fig 4C, its very hard to visually distinguish a remarkable difference between conventional MACE and 3D MACE. Could another image of a different region be provided that shows a marked difference?

2. In the section on Safety,

(a) I note that the MACE was released from the pig only after 7 days post the experiment. Is the delay primarily due to the fact that the presence of the lesion did not allow the capsule to enter the dudenum?

(b) How does the capsule staying within the pig for 7 days, affect the subject? Is there some part of the ethical guidelines of KNOTUS facility which deals with such a situation?

- DISCUSSIONS -

1. Is the difference between MC4000 and MC4000-M, only the presence of the magnetic controller, or is there something else?

2. Regarding the hammer-shaped magnetic controller, pleas refer to my questions in the Methods section about comparing existing magnetic controllers. Ideally please include a comparative imagery of existing handheld magnetic controllers w.r.t to the new dsign?

3. "Therefore, the calculated magnetic forces of the two controllers are similar". The explanations given to explain the force equivalent is unsatisfactory, if dimensions are not provided of conventional controller nor analytical or magnetostatic simulations regarding this.

4. As I mentioned earlier, I do not believe one of the concluding statements "This preclinical trial provides a basis for further clinical studies on 3D MACE." since the submitted manuscript has not detailed a complete GI tract examination and the present study seems to be incomplete and needs further preclinical trials on multiple subjects before becoming a basis for clinical studies.

Reviewer #2: This is an interesting porcine study; there are many typos or mistakes in you submission- please revise.

Furthermore, I would prefer a title like: Gastric examination using a novel three-dimensional (3D) magnetically assisted capsule endoscope and a hand-held magnetic controller: A porcine model study.

Although the majority of the images are useful and of high quality, I believe that the main set up of the experiment, that is (if not a real image) please consider adding an infographic showing the set up of the experiment.

6. PLOS authors have the option to publish the peer review history of their article (what does this mean?). If published, this will include your full peer review and any attached files.

Reviewer #1: No

Reviewer #2: No

---

## [Author Response · Author response to Decision Letter 0]

18 Feb 2021

First of all, thank you very much for giving the opportunity of revision. The authors appreciate the critical and thoughtful comments 

We did our best to write the reply for your comments and to revise the manuscript stating clearly how the text has been changed. And, we added more detailed information and images in the manuscript. 

We indicated precisely where changes have been made in the manuscript, by changing the font to words or sentences on a yellow background. Additional answers were written in underlined with bold type.

[Author’s Answers to Reviewer #1] 

- INTRODUCTION –

1. Please quantify or provide a range in frames-per-sec on what the 'high' in "development of high frame rate of CE camera" refer to.

: We added the sentence based on your comments.

[Page 3, Introduction]

Along with the development of high frame rate of CE camera, some studies reported the use of CE for the evaluation of esophagus and colon (small bowel CE at 2-6 frame rate, esophagus and colon CE at 18-35 frame rate).

- METHODS –

1. This first paragraph describes that "Laboratory animal breeding and experiment were carried out in strict accordance with ethical guidelines and carried out at KNOTUS Co. Ltd (Incheon, Korea), a certified facility." What are the ethical guidelines of KNOTUS? Please attach the guidelines as supplementary information or link it to a publically (publicly) available URL

: It seems that the sentence is somewhat ambiguous. This does not mean that there are guidelines for KNOTUS itself. It means that animal experiment has been carried out by an authorized facility that is registered with the Korea Food & Drug Administration and complies with Korean guideline

(http://www.cdc.go.kr/board.es?mid=a40606010000&bid=0041&act=view&list_no=123031)

: We revised the sentence based on your comments.

[Page 4, Method, 1. Subject]

Laboratory animal breeding and experiment were carried out at animal experiment facility (KNOTUS Co. Ltd, Incheon, Korea) registered with Korea Food & Drug Administration.

2. Why was the study restricted to just one subject? Considering that the MACE was not able to pass the pylorus and travel through the small and large intestines because of the presence of the lesion, isn't this pre-clinical study preliminary and perhaps even incomplete to determine the efficacy of this 3D MACE and new magnetic controller?

: First, it was difficult to pass through pylorus with a gastroscopy due to the lesion. Also, there will be differences in anatomy between human and mini pig (30kg). In humans, normal gut peristalsis alone allows capsule endoscope to pass through pylorus. Besides, in previous human studies using MACE, the capsule easily passed through pylorus by controller. 

: Additional animal experiment using other controllers (the same magnetic force but kettlebell and trackball shape) was conducted in 2021 (KBIO-IACUC-2020-219). In this study, 3D MACE was passed through pylorus. We will attempt the complete stomach and lower GI tract examination through clinical study. 

3. Fig 1A, B, and C are generally of very low quality and appear not in focus. Please replace these images with higher quality images.

(a) - Please use a scale bar and mark the dimensions of the capsule in Fig1A.

: We added the figure based on your comment in “S1 Fig”.

(b) - The pictures of the capsule in Fig1A do not clearly show the 2 camera system that is important detail in this novel capsule. Please replace it with a macro-photograph that showcases the camera and imaging system on the capsule clearly.

: We revised the figure based on your comment. 

(c) Fig C is also of very low quality. None of the text on display nor the label on the product is legible.

: We revised the figure based on your comment. And we added the figure based on your comment in “S3 Fig”.

4. The functionality of the real-time receiver and sensor belt has to be described. What sensors are used in the belt? What kind of receivers on the real-time receiver? Please include a schematic of the whole 3D capsular endoscopy system.

: We added this information at “Method, 2. Devices” section.

[Page 6, Method, 2. Device, (3) Real-time receiver with sensor belt]

Real time receiver uses Android version 9, with an LCD resolution of 800 x 480 pixels. The receiver and sensor belt can be connected. The pads of the sensor belt are composed of 9 conductive fibers (S3 Fig). The images recorded by the 3D MACE are delivered to the real time receiver using a novel transmission technology, known as electrical field propagation. This technology has been applied from the existing small bowel capsule endoscope of the Intromedic Co. Ltd [16].

5.

(a) In Section 2. Devices, Subsection (2) Hand-held magnetic controller, it is not clear what a conventional controller is? Is it an existing Intromedic product? What are its dimensions? How does the new magnetic controller compare to other research-stage or other commercial handheld controllers? Please provide references to other controllers as well. What would be most appropriate is if controllers and their magnet configurations can be compared in a figure.

: The conventional and existing controller is a hammer shape only (not commercial). The hammer shape controller is 26 cm length. So, we studied new controllers (circular-stacked,) with easy-to-manipulate shapes and sizes. We revised the “Fig 1C” based on your comments

[Page 6, Method, 2. Device, (2) Hand-held magnetic controller]

The new controller is smaller in size than an existing controller (hammer shape, 26 cm in length, 6.5cm in hammer head width) [11].

Figure 1. Overview of the three-dimensional (3D) magnetically assisted capsule endoscopy (MACE) system. A: 3D MACE (MiroCam® MC 4000-M), B: circular stacked magnetic controller, C: real-time receiver with sensor belt (MR 2000). The size of the existing controller and the new controller was compared.

(b) It is mentioned that "the magnetic force is equal between the two"? How has this been calculated or estimated. The other sections of the paper do not provide satisfactory explanation to this.

: Although the size of the new controller is smaller than that of the hammer-shaped controller (7 cm versus 26 cm), it uses a high-grade magnet (N54 versus N35). So, the measured magnet force was similar in the new and existing controller.

[Page 6, Method, 2. Device, (2) Hand-held magnetic controller]

But new controller is equipped with a higher-grade magnet than the magnet of existing controller (N54 versus N35). So, the magnetic force measured 10cm away from the controller was similar in the new and existing controller. (30 gauss in new controller and 27 gauss in existing one, respectively).

- RESULTS -

1. In Fig 4C, it’s very hard to visually distinguish a remarkable difference between conventional MACE and 3D MACE. Could another image of a different region be provided that shows a marked difference?

: We revised the figure based on your comments.

2. In the section on Safety,

(a) I note that the MACE was released from the pig only after 7 days post the experiment. Is the delay primarily due to the fact that the presence of the lesion did not allow the capsule to enter the duodenum?

: We put the 3D MACE into the duodenum using the gastroscopic net. So, it took 7 days to release 3D MACE from the small intestine to the anus. Release of 3D MACE was confirmed by experimental facility. We added this information to the manuscript.

[Page 11, Results, 3. Safety]

No significant adverse event was observed when reconfirmed by upper endoscopy. We moved the 3D MACE into the duodenum using the gastroscopic net. The chest radiograph also showed no abnormal findings. The 3D MACE was released from the mini-pig 7 days after the experiment, without any complications.

(b) How does the capsule staying within the pig for 7 days, affect the subject? Is there some part of the ethical guidelines of KNOTUS facility which deals with such a situation?

: After experiment, the subject recovered without any complications and was observed in the experimental facility until the capsule was released. We also checked the breeding records in KNOTUS facility.

- DISCUSSIONS -

1. Is the difference between MC4000 and MC4000-M, only the presence of the magnetic controller, or is there something else?

: The MC4000-M is a new capsule that contains a magnetic inside the MC 4000. There are minor changes in weight and size (0.3 g and 1 mm), but there are no other differences.

2. Regarding the hammer-shaped magnetic controller, please refer to my questions in the Methods section about comparing existing magnetic controllers. Ideally please include a comparative imagery of existing handheld magnetic controllers w.r.t to the new design?

: We added the comparative figure at “Fig 1” based on your comments.

3. "Therefore, the calculated magnetic forces of the two controllers are similar". The explanations given to explain the force equivalent is unsatisfactory, if dimensions are not provided of conventional controller nor analytical or magnetostatic simulations regarding this.

: We measured magnetic force and added this information to the “Method” section.

4. As I mentioned earlier, I do not believe one of the concluding statements "This preclinical trial provides a basis for further clinical studies on 3D MACE." since the submitted manuscript has not detailed a complete GI tract examination and the present study seems to be incomplete and needs further preclinical trials on multiple subjects before becoming a basis for clinical studies.

: Of course, there is still a long way to go for complete GI tract examination. Among the several problems, the active locomotion of capsule endoscope has been solved through many studies. However, measuring the size of lesion and reconstructing stereoscopic images by capsule endoscopy have yet to be studied much. The goal of this study was to develop and verify wireless capsule endoscope that can measure the size of lesions and reconstruct conventional images into 3D images. We will try to solve the remaining problems through further researches.

[Author’s Answers to Reviewer #2] 

1. Furthermore, I would prefer a title like: Gastric examination using a novel three-dimensional (3D) magnetically assisted capsule endoscope and a hand-held magnetic controller: A porcine model study.

: We revised the title based on your comments.

2. Although the majority of the images are useful and of high quality, I believe that the main set up of the experiment, that is (if not a real image) please consider adding an infographic showing the set up of the experiment.

: We added more detailed images in “S4 Fig”

---

## [Decision Letter · Decision Letter 1]

21 Jul 2021

PONE-D-20-31353R1

Gastric examination using a novel three-dimensional (3D) magnetically assisted capsule endoscope and a hand-held magnetic controller: A porcine model study

PLOS ONE

Dear Dr. LIM,

Thank you for submitting your manuscript to PLOS ONE. After careful consideration, we feel that it has merit but does not fully meet PLOS ONE’s publication criteria as it currently stands. Therefore, we invite you to submit a revised version of the manuscript that addresses the points raised during the review process.

We look forward to receiving your revised manuscript.

Kind regards,

Edoardo Sinibaldi

Academic Editor

PLOS ONE

Journal Requirements:

Additional Editor Comments (if provided):

Dear Authors,

although the manuscript was improved, a few further revisions are needed for providing a scientifically sound contribution. In particular, quantitative lesion estimation seems to remain a relatively major point (for which the Reviewer suggests some additional test-bench characterization, also in view of the systematic quest for reduced additional experimental work during the pandemics). Even the supplementary video could provide added value (also based on the overall comments set). Additional points could be tackled through manageable efforts.

Reviewers' comments:

Reviewer's Responses to Questions

**Comments to the Author**

1. If the authors have adequately addressed your comments raised in a previous round of review and you feel that this manuscript is now acceptable for publication, you may indicate that here to bypass the “Comments to the Author” section, enter your conflict of interest statement in the “Confidential to Editor” section, and submit your "Accept" recommendation.

Reviewer #3: All comments have been addressed

2. Is the manuscript technically sound, and do the data support the conclusions?

Reviewer #3: Partly

3. Has the statistical analysis been performed appropriately and rigorously? 

Reviewer #3: N/A

4. Have the authors made all data underlying the findings in their manuscript fully available?

Reviewer #3: Yes

5. Is the manuscript presented in an intelligible fashion and written in standard English?

Reviewer #3: Yes

6. Review Comments to the Author

Reviewer #3:

- Methods 2. Devices (2) Hand-held magnetic controller -

1. Please specify the effective operation distance (in cm) between the controller and the 3D MACE.

2. The authors only mentioned the magnetic flux density (in Gauss) 10 cm away, but not the corresponding magnetic force (in Newton) with respect to the MACE. Please complement this information.

3. If during the operation, the hand-held controller is accidentally moved beyond the effective range with the MACE, would it lost the pose control of MACE? If this is the case, how to re-localize MACE and restore control of MACE? Please comment.

- Methods 3. Porcine gastric examination... -

Please add a picture or a general schematic of the experiment setup, including the test subject (i.e. the mini-pig) together with the arrangement of all the devices.

- Results 2. Verification of images of the gastric... -

The result in terms of lesion size estimation is not quantitatively sound. Eventually only one set of data was reported and assessed. It would be recommended to perform additional test-bench characterization to quantitatively assess the accuracy of the lesion size estimation.

- Supplementary material -

In S2 Fig, different orientations of MACE were shown, however, it would be really interesting to see a supplementary video of the pose (position and orientation) control of MACE under the guidance of the hand-held controller.

7. PLOS authors have the option to publish the peer review history of their article (what does this mean?). If published, this will include your full peer review and any attached files.

Reviewer #3: No

---

## [Author Response · Author response to Decision Letter 1]

3 Aug 2021

First of all, thank you very much for giving the opportunity of revision. The authors appreciate the critical and thoughtful comments 

We did our best to write the reply for your comments and to revise the manuscript stating clearly how the text has been changed. And, we added more detailed information and images in the manuscript. 

[Author’s Answers to Reviewer #3] 

- Methods 2. Devices (2) Hand-held magnetic controller –

1. Please specify the effective operation distance (in cm) between the controller and the 3D MACE. 

: We added the sentence based on your comments. The magnetic force measurement was conducted in the Department of Mechanical Engineering, Hanyang University.

(The effective operation distance between the magnetic controller and 3D MACE is up to 12.5cm.)

2. The authors only mentioned the magnetic flux density (in Gauss) 10 cm away, but not the corresponding magnetic force (in Newton) with respect to the MACE. Please complement this information.

: We complemented this information. The magnetic force measurement was conducted in the Department of Mechanical Engineering, Hanyang University

(The magnetic force with respect to the MACE by the controller was measured to be 41.2 mN at a distance of 10.5 cm.)

3. If during the operation, the hand-held controller is accidentally moved beyond the effective range with the MACE, would it lost the pose control of MACE? If this is the case, how to re-localize MACE and restore control of MACE? Please comment.

: There were no "significant lost control" studies during two additional porcine studies. Even if it was out of effective range, it was not difficult to re-localize the MACE by moving the controller using the real-time viewer.

- Methods 3. Porcine gastric examination... –

Please add a picture or a general schematic of the experiment setup, including the test subject (i.e. the mini-pig) together with the arrangement of all the devices.

: We added actual picture of animal experiment in Supplementary material (S6 Fig)

- Results 2. Verification of images of the gastric... -

The result in terms of lesion size estimation is not quantitatively sound. Eventually only one set of data was reported and assessed. It would be recommended to perform additional test-bench characterization to quantitatively assess the accuracy of the lesion size estimation.

: We conducted a study on object size estimation using Korea Electronics Technology Institute (KETI) test program before animal experiment. In a human intestinal model, polyp models of various sizes were randomly placed. Then, when the size measurement was performed using MACE, the accuracy of 93.37% was confirmed.

- Supplementary material -

In S2 Fig, different orientations of MACE were shown, however, it would be really interesting to see a supplementary video of the pose (position and orientation) control of MACE under the guidance of the hand-held controller.

: We added a short video confirming the movement of MACE under gastroscopy (S5 Video).

---

## [Editor Report · Decision Letter 2]

10 Aug 2021

Gastric examination using a novel three-dimensional magnetically assisted capsule endoscope and a hand-held magnetic controller: A porcine model study.

PONE-D-20-31353R2

Dear Dr. LIM,

We’re pleased to inform you that your manuscript has been judged scientifically suitable for publication and will be formally accepted for publication once it meets all outstanding technical requirements.

Kind regards,

Edoardo Sinibaldi

Academic Editor

PLOS ONE

Additional Editor Comments (optional):

The manuscript was improved by leveraging the raised comments.
---

## [Editor Report · Acceptance letter]

27 Sep 2021

PONE-D-20-31353R2 

Gastric examination using a novel three-dimensional magnetically assisted capsule endoscope and a hand-held magnetic controller: A porcine model study. 

Dear Dr. Lim:

I'm pleased to inform you that your manuscript has been deemed suitable for publication in PLOS ONE. Congratulations! Your manuscript is now with our production department. 

Kind regards, 

on behalf of

Dr. Edoardo Sinibaldi 

Academic Editor

PLOS ONE